# The Association between Vegan, Vegetarian, and Omnivore Diet Quality and Depressive Symptoms in Adults: A Cross-Sectional Study

**DOI:** 10.3390/ijerph20043258

**Published:** 2023-02-13

**Authors:** Hayley Walsh, Megan Lee, Talitha Best

**Affiliations:** 1Gold Coast Campus, Bond University, Robina, QLD 4226, Australia; 2NeuroHealth Lab, Appleton Institute, School of Health, Medical and Applied Science, Central Queensland University, Brisbane, QLD 4000, Australia

**Keywords:** dietary pattern, depression, plant-based, meat-based, diet quality

## Abstract

Dietary patterns and depressive symptoms are associated in cross-sectional and prospective-designed research. However, limited research has considered depression risk related to meat-based and plant-based dietary patterns. This study explores the association between diet quality and depressive symptoms across omnivore, vegan, and vegetarian dietary patterns. A cross-sectional online survey utilised the Dietary Screening Tool (DST) and the Centre for Epidemiological Studies of Depression Scale (CESD-20) to measure diet quality and depressive symptoms, respectively. A total of 496 participants identified as either omnivores (*n* = 129), vegetarians (*n* = 151), or vegans (*n* = 216). ANOVA with Bonferroni post hoc corrections indicates that dietary quality was significantly different between groups F(2, 493) = 23.61, *p* < 0.001 for omnivores and vegetarians and omnivores and vegans. Diet quality was highest in the vegan sample, followed by vegetarian and omnivore patterns. The results show a significant, moderately negative relationship between higher diet quality and lower depressive symptoms (r = −0.385, *p* < 0.001) across groups. Hierarchical regression showed that diet quality accounted for 13% of the variability in depressive symptoms for the omnivore sample, 6% for vegetarians, and 8% for vegans. This study suggests that diet quality in a meat-based or plant-based diet could be a modifiable lifestyle factor with the potential to reduce the risk of depressive symptoms. The study indicates a greater protective role of a high-quality plant-based diet and lower depressive symptoms. Further intervention research is needed to understand the bi-directional relationship between diet quality and depressive symptoms across dietary patterns.

## 1. Introduction

The cost of depression to the global economy is an estimated US$1 trillion annually, impacting 5% of the global adult population [1] and 10% of Australians [2]. This chronic disorder is characterised by mood dysregulation, sad, empty, or irritable emotions, as well as negative self-appraisal and withdrawal/isolating behaviours [3]. Depression is responsible for 50–70% of suicides globally and is the second leading cause of death for 15–29 year-olds [4]. The average treatment response rate for depression is 20–30% [5], prompting research to consider other modifiable lifestyle factors, such as diet, to address depressive symptoms [6].

The emerging field of nutritional psychiatry is the nexus between nutrition and psychology, focusing on the role of dietary patterns in mental health conditions such as depression [6,7,8]. Dietary patterns, defined as “the quantity, variety, or combination of different foods and beverages in a diet and the frequency with which they are habitually consumed” [9], are theorised to impact mood due to the differing nutrient profiles and biological mechanisms [10]. Dietary patterns are categorised as either ‘healthy’—rich in fresh vegetables, fruits, seeds, nuts, whole grains, legumes, and water or ‘unhealthy’—high in refined, sugary, and ultra-processed foods [11]. The most commonly researched healthy dietary pattern—the Mediterranean dietary pattern—consists of high consumption of fruit, vegetables, nuts, and olive oil, moderate consumption of oily fish, and limited intake of red meat and highly processed foods [12]. Adherence to the Mediterranean diet has been associated with a lower risk of depression onset [13,14], whilst consumption of a Western dietary pattern typically includes high consumption of processed foods, meat, dairy, and alcohol [15] and is associated with increased risk of depression [16].

Many observational, longitudinal, and intervention studies exploring diet and depression primarily focus on healthy (Mediterranean, anti-inflammatory) and unhealthy (Western) dietary patterns [6,17,18,19,20,21,22,23]. Four randomised control trials assessed dietary change from unhealthy (Western) to healthy (Mediterranean) dietary patterns and depression; two assessed the general population [14,18,24] and two assessed young adults [19,20]. All four studies found that the depressive symptoms of the participants significantly improved after the healthy dietary intervention compared with the control group.

To date, associations between dietary patterns and depression outcomes are diverse and exist across the lifespan, including childhood through to older adulthood [25]. For example, narrative systematic review findings of 20 studies in children and young adults show that high diet quality is associated with lower levels of depression. Conversely, low-quality diets are associated with higher levels of depression [26]. Additionally, a systematic review and meta-analysis of 18 studies on dietary patterns and depression risk in older adults found that high diet quality was associated with lower depression risk (OR, 0.85; 95%CI, 0.78–0.92) [27]. Other systematic reviews and meta-analyses in the general population have found similar findings [25,28,29]. However, high levels of heterogeneity and risk of bias are inherent across these findings [30,31]. More research is needed in young adult populations to support awareness of dietary patterns and depressive symptoms.

Dietary patterns can be further categorised into omnivore and/or plant-based. The typical omnivore diet includes no restrictions on animal products and is generally high in arachidonic acid, a fatty acid found in meat and linked to lower mood [32,33]. Plant-based dietary patterns are characterised by their emphasis on fruits, vegetables, whole grains, soy foods, nuts, and seeds; whilst a vegan diet excludes all animal products, a vegetarian diet may include dairy and eggs [34]. Importantly, low-quality foods such as those high in sugar, saturated fats, and refined grain consumption are also consumed in plant-based dietary patterns. Therefore, both meat and plant-based diets have the potential to be high or low in diet quality [35].

The relationship between dietary patterns, predominately plant-based vegetarian and vegan patterns, and depression is equivocal. Some research suggests that vegetarians and vegans have increased depressive symptoms compared to their omnivore counterparts [36]. For example, meat abstinence is associated with depressive symptoms and, when compared to other dietary patterns, meat abstainers exhibit more significant symptoms of depression than meat eaters [37,38,39]. Conversely, other studies demonstrate lower symptoms of depression in plant-based diet samples than in omnivore samples [33,40,41,42]. Systematic reviews report inconsistent findings between a plant-based dietary pattern and depressive symptoms [43,44]. These inconsistent findings suggest that it may not be plant-based dietary patterns that are linked to depression but rather the quality of the plant-based diet. Additionally, the association between plant-based diets and depression may be confounded by other factors that impact mental health, such as food restriction and food group exclusion, despite the absence of meat [45]. Therefore, more research is needed to understand the relationship between plant-based dietary patterns and depressive symptoms.

A recent Australian study surveyed 219 vegans and vegetarians aged 18 to 44 years and considered a plant-based diet quality measurement. Results showed that a high-quality plant-based diet was associated with reduced depressive symptoms, and a low-quality plant-based diet was associated with increased depressive symptoms [42]. Whilst this finding aligns with high and low-quality diet impacts on health in the general population [17], the study focused solely on the relationship between a plant-based diet quality measure and depressive symptoms. As such, further diet quality comparison with omnivore diets is needed.

The primary objective of this research is to extend the previous study and compare diet quality in vegan, vegetarian, and omnivore populations and the association with depressive symptoms. It is hypothesised that high diet quality will be associated with decreased depressive symptoms and that there will be a difference in diet quality and depressive symptoms between the vegan, vegetarian, and omnivore populations.

## 2. Material and Methods

A total of 581 participants started the survey. The complete data response rate was 85% (*n* = 496) for the primary outcome variables (diet quality and depressive symptoms). A G Power analysis indicated that 219 participants would be adequate for a moderate effect (0.15, α = 0.05, β = 0.80). Participants were recruited online via social media sites Facebook, Twitter, or SONA (Bond University student research portal). Inclusion criteria required English-speaking participants between 18 and 44 years of age to have access to an internet-enabled device. Participants were informed that no remuneration was provided for participation, excluding students who accessed the survey through the SONA platform for course credit. The research team first piloted the survey to ensure usability. Informed consent was granted after participants read the information sheet and commenced the survey. Due to a large influx of vegan participants, an attempt to acquire representation from omnivores and vegetarians was made through purposive sampling.

Data collection occurred between November 2021 and January 2022. The Strengthening the Reporting of Observational Studies in Epidemiology (STROBE-nut) checklist ensured a common reporting standard Appendix A [46]. The Bond University Human Research Ethics Committee approved the study (#ML01980). Participants completed an online survey hosted on Qualtrics, which asked for demographic measures of gender (male, female), age (continuous), marital status (partnered, not partnered), and education (high school, university degree, trade certificate). Self-reported height and weight were collected to calculate the participants’ Body Mass Index (BMI). Participants self-reported dietary patterns (omnivore, vegan, or vegetarian).

### 2.1. Depression (CESD-20)

The CESD-20 [47] is a 20-item measurement of symptoms of depression. The scale indicates the experience of depressive symptoms in the general population, not in a clinical population. Depressive symptoms experienced within the previous seven days were calculated on a four-point Likert scale: rarely (0) to most of the time (3). An example question included: “I felt I was just as good as other people”. Scores ranged between 0 and 60; the higher the score, the greater the experience of depressive symptoms. A score of 16 or above exceeded the criterion cut-off score and indicated the experience of depressive symptoms. The CESD-20 has been successfully employed on younger populations, older populations, and populations with health comorbidities [48]. The CESD-20 demonstrated high validity, reliability [49], and internal consistency (α > 0.9). The tool is concurrently valid with the Beck Depression scale, classified as the gold standard scale for depression measurement [48].

### 2.2. Dietary Screening Tool (DST)

The DST determines diet quality and nutritional risk scores by capturing the frequency of commonly eaten foods over a seven-day period [50]. The original American scale consists of 37 items and is structured according to the 2005 Dietary Guidelines for Americans [50]. The DST version used in this study was condensed to 21 items, with adaptations for an Australian population, which have been used in previous research [14]. For example, the fast-food question was amended to include Australian fast-food chains: “How often do you eat McDonald’s, Kentucky Fried Chicken, Pizza Hut, or Hungry Jacks?” Each food item response was assigned a number between 0 and 8. The total sum of all scores determined diet quality, with 0 indicating low diet quality and 105 signifying high diet quality. The DST had a high test-retest reliability coefficient of 0.83 (*p* < 0.001) and high validity and was validated in an Australian population [50,51].

### 2.3. International Physical Activity Questionnaire (IPAQ)

The IPAQ [52] determined physical activity levels over a seven-day period, capturing activity as vigorous, moderate, walking, or sitting across 7 items, for example: “During the last 7 days, on how many days did you do vigorous physical activities like heavy lifting, digging, aerobics, or fast bicycling?” For the purpose of this study, total active hours per week were calculated by tallying vigorous, moderate, and walking activities. The IPAQ had a high test-retest reliability coefficient of 0.80 (*p* < 0.001) and high validity [53].

### 2.4. The Social Connectedness Scale—Revised

The Social Connectedness Scale [54] determines the degree to which participants have felt connections to others in social settings. Participants provide a response to eight items on a six-point Likert scale, ranging from strongly disagree (1) to strongly agree (6). An example question includes: “I feel so distant from people”. Negatively worded items were reverse-coded. Scores were summed to provide total scores (range 0–48), with higher scores indicating a strong sense of social connectedness, high reliability (internal consistency a >0.92), and validity [55].

### 2.5. Statistical Analysis

Data were analysed using SPSS Statistics software version 28 (IBM SPSS Statistics, Chicago, FL, USA). Steps were taken to ensure data integrity; missing values in the CESD-20 and DST were imputed with mean estimates based on the dietary pattern category. Frequency descriptives were calculated for all variables prior to ANOVA and chi-square statistics were conducted to compare group means. The final ANOVA analysis included a Bonferroni correction post hoc in determining between-group differences, segmented by dietary pattern, for all variables. Correlational analyses were used to determine the association between depressive symptoms (DV), diet quality (IV), and other covarying factors such as age, gender, BMI, social connectedness, physical activity levels, marital status, and education level. Pearson’s correlation was used to assess the significance of continuous variables and Spearman’s correlation for categorical variables. Subsequently, a hierarchical multiple linear regression was conducted for all significant continuous predictors of depressive symptoms (CESD-20); BMI and physical activity levels for model one and DST for model two.

## 3. Results

The demographic and lifestyle characteristics of 496 participants are detailed in Table 1, split by dietary pattern and total population. Overall, the participants’ mean age was 30.95 years (*SD* = 7.47), with 76% identifying as female. Most were partnered (66%) and had a university degree (75%). The overall population was split by dietary pattern: omnivore (*n* = 129), vegetarian (*n* = 151), and vegan (*n* = 216).

The mean depressive symptom score (CESD-20) for participants following an omnivore dietary pattern was 16.27 (*SD* = 10.98). The mean omnivore score is above the cut-off criterion score of 16, indicating possible experiences of depressive symptoms for this dietary pattern. Depressive symptom scores for those following a vegetarian dietary pattern (*M* = 12.99, *SD* = 9.78) and vegan dietary pattern (*M* = 11.08, *SD* = 9.83) were below the criterion cut-offs as were the total population scores (*M* = 13.01, *SD* = 10.32). A one-way ANOVA found a significant difference in diet quality, *F*(2, 493) = 23.61, *p* < 0.001, and depressive symptoms, *F*(2, 493) = 10.61, *p* < 0.001 between the three dietary patterns (omnivore, vegetarian, and vegan). A post hoc analysis with Bonferroni correction determined that vegans (*M* = 76.55, *SD* = 10.44, *p* < 0.001) and vegetarians (*M* = 73.00, *SD* = 12.00, *p* < 0.001) had greater dietary quality than omnivores (*M* = 67.03, *SD* = 15.66).

In the overall sample, Pearson’s correlations on continuous variables and Spearman’s correlations on categorical variables (Table 2) show a significant moderate negative relationship between diet quality and depressive symptoms (*r* = −0.385, *p* < 0.001), irrespective of dietary type. For dietary type, the significant moderately negative relationship between diet quality and depressive symptoms remained, see Table 3, omnivore (*r* = −0.440, *p* < 0.001), vegetarian (*r* = −0.302, *p* < 0.001), and vegan (*r* = −0.300, *p* < 0.001).

A two-stage hierarchical multiple linear regression was conducted to estimate the proportion of variance in depressive symptoms that can be accounted for by BMI, physical activity levels, and diet quality across omnivore, vegetarian, and vegan dietary patterns. Standardised (β) regression coefficients and squared semi-partial correlations (sr^2^) for each predictor at each step are reported in Table 4. Prior to interpreting the results, the assumptions of linearity, homoscedasticity, and multicollinearity were met.

On the first step of the hierarchical multiple linear regression, model one, BMI and physical activity levels collectively accounted for 9% of the variability in depressive symptoms in omnivores, *R*^2^ = 0.094, *F*(2, 126) = 6.57, *p* = 0.002. Model one accounted for 8% of depressive symptoms in vegetarians, *R*^2^ = 0.078, *F*(2, 148) = 6.25, *p* = 0.002; however, no significant results were determined for vegans (*p* = 0.079). In the second step of the model, diet quality was added to the regression equation and accounted for an additional 13% of the variability in depression symptoms for an omnivore diet, *R*^2^ = 0.229, *F*(1, 125) = 21.8, *p* < 0.001. Model two accounted for an additional 6% of the variability in depression symptoms for a vegetarian diet *R*^2^ = 0.134, *F*(1, 147) = 9.45, *p* < 0.001, and 8% for a vegan diet, *R^2^* = 0.100, *F*(1, 212) = 18.06, *p* < 0.001.

## 4. Discussion

The current study explored the relationship between diet quality and symptoms of depression in self-reported vegan, vegetarian, and omnivore dietary samples. Across the whole sample of dietary patterns, a high-quality diet was associated with lower depressive symptoms. These findings align with recent research that showed a high-quality plant-based diet may protect against the onset or severity of depression [42] and that improved dietary quality, in general, is related to lowering the symptoms of depression [17,56,57,58].

Notably, the relationship between dietary quality and reported depressive symptoms was irrespective of dietary patterns and reflects the broad findings from intervention studies. For example, four Australian randomised controlled trials show that by increasing dietary quality, symptoms of depression decreased over a 12-week time frame [14,18,19,20]. Further, systematic review and meta-analysis findings show that adhering to a high-quality diet protects against depressive symptoms [17]. Whilst the relationships between high dietary quality and lower depression appear robust, the type of dietary pattern has not been considered, and the neurophysiological mechanisms of effect for differential dietary patterns and relationship with depression are yet to be clearly understood. Our results show that across and between dietary patterns, diet quality relates to depressive symptoms, irrespective of dietary pattern.

Probable explanations for the relationship between diet quality and depressive symptoms rely on biological mechanisms underpinning the biochemical effects of specific food components in a low-quality diet. For example, a low-quality diet is typically characterised by ultra-processed foods [59] or those constituting high sugar or fat content [28]. Consumption of high-sugar and high-fat foods is associated with heightened markers of inflammation and inflammatory disease [60], which in turn are associated with a higher risk of depression [61,62]. Some low-quality foods, including sugar-sweetened beverages, have a high glycaemic index that also contributes to inflammation [63]. In individuals with depression, higher inflammation markers are present compared to healthy control groups, suggesting that biological mechanisms of inflammation underpin depression [64]. Similarly, higher oxidative stress and inflammation levels have been identified in the brains of those with depression than in those without depression [65].

To date, dietary interventions such as the Mediterranean diet (high in plant foods) are known to reduce inflammation and inflammatory markers [66,67] and reduce depressive symptoms [13,18]. Similarly, diets high in fruit and vegetables, which are staples of a vegan or vegetarian diet, are rich in antioxidants, notably polyphenols, which are negatively correlated with depression [68,69] and depression severity [65].

Conversely, nutritional deficiencies in a plant-based diet could be involved in the increase of depressive symptoms. One study showed that 52% of vegans and 7% of vegetarians were deficient in vitamin B12 [70], a vitamin generally gained through red meat consumption and thought to help combat depressive symptoms [38]. Similarly, omega-3 polyunsaturated fatty acids play a vital role in brain function and are linked with mood outcomes [33]. Given that the most bioavailable source of Omega-3 polyunsaturated fatty acids is oily fish, the intake of fish is reduced in some plant-based diets, and as such omega-3 deficiency may play a role in decreased mood [44].

When split by dietary pattern, the current findings reveal that the vegan sample reported the highest diet quality score, followed by the vegetarian and omnivore samples. These findings are consistent with previous research that reported that both vegan and vegetarian dieters scored higher on the Healthy Eating Index 2010 (HEI-2010) than omnivores (Clarys, et al. [71]). Results are likely attributed to the nutritional value of foods consumed in each dietary pattern. For example, a vegan diet typically includes more whole foods, such as vegetables and legumes. In contrast, an omnivore diet typically includes more ultra-processed and refined foods such as cake, pastries, and chocolate [72]. When exceeding recommended amounts, sugar, sodium, and fat levels are associated with a low-quality diet and are considerably higher in self-nominated omnivore foods [73]. Conversely, a more recent study showed that both vegetarian and vegan dietary patterns of a low-quality diet were associated with high levels of depression compared to a high-quality vegan or vegetarian diet [42]. These studies suggest that diet quality may be associated with depressive symptoms, irrespective of the dietary pattern consumed.

In this study, the omnivore diet quality score was significantly lower than both the vegetarian and vegan groups and may result from the frequency of consumption of red and processed meat and lower consumption of fruit and vegetables compared to their vegan and vegetarian counterparts. Red meat is commonly touted as a valuable contributor to the recommended dietary intake levels of protein, iron, vitamin B12, and zinc [74], all of which contribute to a healthy diet. However, excess consumption can result in the digestion of too many saturated fats, which is associated with a low-quality diet and a higher risk of depressive symptoms [75]. It has been reported that a typical omnivore consumes approximately 58% and 81% more meat than the recommended intake for women and men, which correspondingly [76] links excess red meat consumption with lower diet quality in comparison to plant-based diets. In the first study to look at meat consumption in depression Jacka, Pasco, Williams, Mann, Hodge, Brazionis and Berk [77] found that when consumption rates fell below or exceeded the recommended daily intake (28–57 g per day), red meat consumption was linked to an increased risk of depressive disorder prevalence. These findings highlight that both abstinence from red meat and overconsumption may have adverse implications for mood.

Depressive symptoms were highest in the omnivore group in this study, followed by the vegetarian and vegan groups. The correlation between diet quality and depressive symptoms between groups followed a similar trend, as the strongest relationship was evident in the omnivore sample, followed by the vegetarian and then vegan samples. Results show a significant between-group difference when comparing the omnivore and vegan sample and the omnivore and vegetarian sample; however, no difference was found between the vegan and vegetarian sample. Literature on plant-based diets and depressive symptoms is ambiguous. Some studies found that plant-based diets were associated with a risk of depressive symptoms [42,70]. Other studies only concluded gender differences; male vegetarians demonstrated higher depression than male omnivores, but females did not [38]. Interestingly, a recent meta-analysis of over 170,000 participants concluded that people who eat meat, predominantly omnivores, had lower depression than plant-based diet samples [78].

### Strengths, Limitations, and Future Direction

The study is novel to the field, providing a unique insight into the relationship between diet quality and depressive symptoms across vegan, vegetarian, and omnivore dietary patterns. Key strengths of the study include the adequate power of the sample size and high completion rate (79%), which indicates a highly motivated sample. A sensitivity analysis was conducted to explore the impact of the three DST items that scored favourably on the omnivore dietary pattern, determining no disadvantage to the plant-based diet sample and further endorsing the study’s findings.

Conversely, the study’s highly motivated participants are also a limitation as the sample may not reflect the general Australian population who have a predominantly Western dietary pattern. Further, in this sample, 61% of participants were within the self-reported healthy BMI of 25 compared to the national average of 32% [79]. Data collection for this study occurred during the COVID-19 worldwide pandemic. As such, it would be expected that data may not reflect habitual dietary patterns due to the global increase in the consumption of high-energy-density snack foods and emotional eating [80]. In addition, global depression rates had also increased during the pandemic and may have impacted the results of this study [81]. Further, selection bias may be a concern as participation in the study may be driven by following a plant-based diet or experiencing depressive symptoms. This may explain why omnivore participants reported higher levels of depressive symptoms.

Some of the recruitment occurred on a plant-based social media site and a podcast in which a study reflecting poorly on plant-based diets was discussed. Therefore, a proportion of the vegan and vegetarian responses to the CESD-20 depression scale in this study may have been influenced by social desirability. Therefore, depressive symptoms being highest in the omnivore diet, in comparison to the plant-based diet samples, may be influenced by this social desirability bias. Being a morally motivated minority, vegan groups are often stigmatised and the target of discrimination [82] and may inadvertently mask depressive symptoms to maintain socially desirable attributes of group identity. Therefore, comparisons between meat-based and plant-based diet followers and depressive symptoms within this study should be interpreted with due prudence.

## 5. Conclusions

This cross-sectional study demonstrates an association between high-quality dietary omnivore, vegan, and vegetarian diets and lower depressive symptoms. It also indicates that vegan and vegetarian dietary patterns were associated with higher diet quality. Regardless of the dietary pattern (meat or plant-based), the importance of these findings shows that more frequent intake of fruits, vegetables, nuts, seeds, legumes, whole grains, and water and reduced ultra-processed, refined, and sugary foods, are associated with lower depressive symptoms.

## Figures and Tables

**Table 1 ijerph-20-03258-t001:** Demographic and lifestyle statistics per dietary pattern.

Dietary Pattern	Omnivore (*n* = 129)	Vegetarian (*n* = 151)	Vegan (*n* = 216)	Full Sample (*n* = 496)	Test Statistic
Gender (*n*, %)					Χ^2^ = 14.14 *
Female	96 (74)	127 (84)	152 (70)	375 (76)	
Male	33 (26)	23 (15)	62 (29)	118 (24)	
Marital status (*n*, %)					Χ^2^ = 1.01
Partnered	86 (67)	104 (69)	138 (64)	328 (66)	
Unpartnered	43 (33)	47 (31)	78 (36)	168 (34)	
Highest educational level (*n*, %)					Χ^2^ = 9.94 *
High school	33 (26)	32 (21)	29 (13)	94 (19)	
University degree	86 (67)	112 (74)	173 (80)	371 (75)	
Trade certificate	10 (8)	7 (5)	14 (7)	31 (6)	
Age (years; *M*, *SD*)	31.28 (8.07)	30.52 (7.80)	31.04 (6.86)	30.95 (7.47)	*F* = 0.39
BMI ^d^ (kg/m^2^; *M*, *SD*)	25.66 ^a^ (5.68)	23.72 ^bc^ (4.94)	23.27 ^cb^ (4.61)	24.03 (5.09)	*F* = 9.59 **
DST ^e^ (*M*, *SD*)	67.03 ^a^ (15.66)	73.00 ^bc^ (12.00)	76.55 ^cb^ (10.44)	72.99 (13.01)	*F = 23.61 ***
IPAQ ^f^ (hours per week; *M*, *SD*)	9.72 ^ab^ (12.15)	10.18 ^bac^ (10.59)	12.74 ^cb^ (13.30)	11.18 (12.29)	*F* = 3.18 *
CESD-20 ^g^ (*M*, *SD*)	16.27 ^a^ (10.98)	12.99 ^bc^ (9.78)	11.08 ^cb^ (9.83)	13.01 (10.32)	*F* = 10.61 **
DASS-21 Depression ^h^ (*M*, *SD*)	4.79 ^ab^ (4.59)	4.08 ^bac^ (4.15)	3.60 ^cb^ (4.17)	4.06 (4.30)	*F = 3.08 **
DASS-21 Anxiety ^i^ (*M*, *SD*)	4.33 ^a^ (3.57)	3.25 ^b^ (3.61)	2.37 ^c^ (2.86)	3.14 (3.38)	*F = 14.08 ***
SCS ^j^ (*M*, *SD*)	84.44 (18.34)	85.60 (17.81)	83.30 (18.30)	84.28 (18.15)	*F = 0.68*

Note: * *p* < 0.05, ** *p* < 0.001; ^a–c^ different letters in the same row indicate significant statistical difference (*p* < 0.05) as indicated by Bonferroni correction; ^d^ BMI, Body Mass Index; ^e^ DST, Dietary Screening Tool; ^f^ IPAQ, International Physical Activity Questionnaire; ^g^ CESD-20, Centre for Epidemiological Studies Depression; ^h^ DASS Depression, Depression Anxiety Stress Scale—Depression; ^i^ DASS Anxiety, Depression Anxiety Stress Scale—Anxiety; ^j^ SCS, Social Connectedness Scale; *M*, Mean; *SD*, Standard Deviation.

**Table 2 ijerph-20-03258-t002:** Correlation matrix of overall population.

	CESD-20 ^a^	DST ^b^	BMI ^c^	Age	Gender	Marital Status	IPAQ ^d^	Education Level	SCS ^e^	DASS-Depression ^f^	DASS-Anxiety ^g^
DST ^b^	−0.385 **										
BMI ^c^	0.201 **	−0.268 **									
Age	−0.132 **	0.117 **	0.111 *								
Gender											
Marital Status					−0.042 ^h^						
IPAQ ^d^	−0.167 **	0.190 **	−0.127 **	0.001							
Education level					−0.017 ^h^	−0.125 ** ^h^					
SCS ^e^	−0.610 **	0.198 **	−0.154 **	0.040			0.097 *				
DASS-Depression ^f^	0.868 **	−0.314 **	0.127 **	−0.086			−0–0.141 **		−0.589 **		
DASS-Anxiety ^g^	0.663 **	−0.306 **	0.195 **	−0.143 **			−0.132 **		−0.404 **	0.609 **	

Note: * *p* < 0.05, ** *p* < 0.001; ^a^ CESD-20, Centre for Epidemiological Studies Depression; ^b^ DST, Dietary Screening Tool; ^c^ BMI, Body Mass Index; ^d^ IPAQ, International Physical Activity Questionnaire; ^e^ SCS, Social Connectedness Scale; ^f^ DASS-Depression, Depression Anxiety Stress Scale—Depression; ^g^ DASS-Anxiety, Depression Anxiety Stress Scale—Anxiety; ^h^ indicates Spearman’s correlation for categorical variables.

**Table 3 ijerph-20-03258-t003:** Pearson’s correlation of diet quality and depressive symptoms split by dietary pattern.

Omnivore	−0.440 **
Vegetarian	−0.302 **
Vegan	−0.300 **

Note: ** *p* < 0.001.

**Table 4 ijerph-20-03258-t004:** Linear regression between an omnivore, vegetarian, and vegan sample.

	Model		β	*t*	sr^2^	*R*	*R* ^2^	*Adj R* ^2^
Omnivore	1	(Constant)		1.438		0.307	0.094	0.080
		BMI ^b^	0.231	2.716 *	0.230			
		IPAQ ^c^	−0.183	−2.148 *	−0.182			
	2	(Constant)		4.570 **		0.478	0.229	0.210
		BMI ^b^	0.126	1.540	0.121			
		IPAQ ^c^	−0.138	−1.743	−0.137			
		DST ^d^	−0.385	−4.666 **	−0.366			
Vegetarian	1	(Constant)		2.808 *		0.279	0.078	0.065
		BMI ^b^	0.093	1.180	0.093			
		IPAQ ^c^	−0.256	−3.241 **	−0.256			
	2	(Constant)		4.172 **		0.365	0.134	0.116
		BMI ^b^	0.039	0.498	0.038			
		IPAQ ^c^	−0.206	−2.620 *	−0.201			
		DST ^d^	−0.248	−3.073 *	−0.236			
Vegan	1	(Constant)		1.181		0.154	0.024	0.014
		BMI ^b^	0.144	2.097 *	0.142			
		IPAQ ^c^	−0.036	−0.529	−0.036			
	2	(Constant)		4.211 **		0.317	0.100	0.088
		BMI ^b^	0.104	1.569	0.102			
		IPAQ ^c^	0.007	0.102	0.007			
		DST ^d^	−0.284	−4.250 **	−0.277			

Note: ^b^ BMI, Body Mass Index; ^c^ IPAQ, International Physical Activity Questionnaire; ^d^ DST, Dietary Screening Tool, * *p* < 0.05, ** *p* < 0.001.

## Data Availability

Data is available at https://data.mendeley.com/datasets/pjnmhf4sx5/1 (accessed on 10 January 2022).

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
