# Peer review of "The Association between Vegan, Vegetarian, and Omnivore Diet Quality and Depressive Symptoms in Adults: A Cross-Sectional Study"

_ijerph, 2023, doi:10.3390/ijerph20043258_

Round 1
Reviewer 1 Report
Dear Authors,
Thanks for this interesting and well-structured research.
Abstract: Sufficient
Introduction:
Line 44-52 The authors stated that "Dietary patterns are categorised as either healthy or unhealthy. The most common healthy dietary pattern is the Mediterranean dietary pattern.” Vegan, vegetarian, and omnivore diet forms can be mentioned here. Also, the relationship between these dietary patterns and depression can be mentioned. The following studies may be helpful.
1. Gianfredi, V., Dinu, M., Nucci, D., Eussen, S. J. P. M., Amerio, A., Schram, M. T., Schaper, N., & Odone, A. (2022). Association between dietary patterns and depression: an umbrella review of meta-analyses of observational studies and intervention trials. Nutrition Reviews, Advance online publication. https://doi.org/10.1093/nutrit/nuac058
2. Li, Y., Lv, M. R., Wei, Y. J., Sun, L., Zhang, J. X., Zhang, H. G., & Li, B. (2017). Dietary patterns and depression risk: A meta-analysis. Psychiatry Research, 253, 373-382.
Line 53-71 makes it hard to follow for the reader. Therefore, it should be removed.
Line 81-83 “The relationship between plant-based diets and depression is equivocal.” Dietary patterns should be called instead of plant-based diets. The emphasis on vegan, vegetarian and omnivorous diets should be clearer.
Line 103 “The primary objective of this research is to replicate the previous study”. I did not understand what was meant by the replicate the previous study.
Line 105-107 “It is hypothesised that high diet quality will be associated with decreased depressive symptoms, and low diet quality will be associated with increased depressive symptoms across groups.” There is no need to repeat the opposite sentences. Also, this sentence should be removed.
Material and Methods: Sufficient
Discussion
Line 224-226: “Across the whole sample of dietary patterns, ……….”. “and a low-quality diet was associated with higher depressive symptoms” should be removed.
Author Response
Thank you for taking the time to review our manuscript.

Reviewer 2 Report
Please see the attached file.

Author Response
Thank you for taking the time to review our manuscript
